# Tailoring the Emission Behavior of WO_3_ Thin Films by Eu^3+^ Ions for Light-Emitting Applications

**DOI:** 10.3390/nano13010007

**Published:** 2022-12-20

**Authors:** V. S. Kavitha, V. Biju, K. G. Gopchandran, R. Praveena, C. K. Jayasankar, Wanichaya Mekprasart, Kanokthip Boonyarattanakalin, Wisanu Pecharapa, V. P. Mahadevan Pillai

**Affiliations:** 1Department of Optoelectronics, University of Kerala, Kariavattom, Thiruvananthapuram 695581, India; 2Department of Nanoscience and Nanotechnology, University of Kerala, Kariavattom, Thiruvananthapuram 695581, India; 3Department of Physics, Gayatri Vidhya Parishad College of Engineering (A), Visakhapatnam 530048, India; 4Department of Physics, Sri Venkateswara University, Tirupati 517502, India; 5College of Materials Innovation and Technology, King Mongkut’s Institute of Technology Ladkrabang, Bangkok 10520, Thailand; 6Amrita School of Physical Science, Amrita VishwaVidyapeetam, Amaravathi Campus, Coimbatore 641112, India

**Keywords:** films, defects, optical properties, Burstein–Moss effect

## Abstract

The article reports the successful fabrication of Eu^3+^-doped WO_3_ thin films via the radio-frequency magnetron sputtering (RFMS) technique. To our knowledge, this is the first study showing the tunable visible emission (blue to bluish red) from a WO_3_:Eu^3+^ thin film system using RFMS. X-ray diffractograms revealed that the crystalline nature of these thin films increased upto 3 wt% of the Eu^3+^ concentration. The diffraction peaks in the crystalline films are matched well with the monoclinic crystalline phase of WO_3_, but for all the films’, micro-Raman spectra detected bands related to WO_3_ monoclinic phase. Vibrational and surface studies reveal the amorphous/semi-crystalline behavior of the 10 wt% Eu^3+^-doped sample. Valence state determination shows the trivalent state of Eu ions in doped films. In the 400–900 nm regions, the fabricated thin films show an average optical transparency of ~51–85%. Moreover, the band gap energy gradually reduces from 2.95 to 2.49 eV, with an enhancement of the Eu^3+^-doping content. The doped films, except the one at a higher doping concentration (10 wt%), show unique emissions of Eu^3+^ ions, besides the band edge emission of WO_3_. With an enhancement of the Eu^3+^ content, the concentration quenching process of the Eu^3+^ ions’ emission intensities is visible. The variation in CIE chromaticity coordinates suggest that the overall emission color can be altered from blue to bluish red by changing the Eu^3+^ ion concentration.

## 1. Introduction

Tungsten oxide, referred to as WO_3_, is a well-known n-type wide-energy band gap oxide semiconductor material that has gained much attention as a promising candidate for different applications in various fields [1]. It shows exciting novel functionalities because of its unusual electronic behavior. Generally, different nanostructures of WO_3_ such as nanorods, nanoparticles, nanowires, nanoplates, etc., are of great concern as a promising candidate for electrochromic applications [2,3], photocatalysis [4], gas sensors [5,6], solar cells [7], etc. The general morphology of WO_3_ is defect-perovskite, with the formula of ABO_3_. In the ABO_3_ structure, the ‘A’ sites are unoccupied, whereas the ‘B’ sites are occupied with ‘W’ atoms, forming a 3-D network of corner or edge-shared or a layered structure of WO_6_ octahedra, depending on whether they are stoichiometric WO_3_ or sub-stoichiometric WO_3_ or WO_3_ hydrates [1]. Thin films of WO_3_ have been used as gas sensors [8] and the photocatalyst [9] in photochromic devices [10] and as transparent conducting electrodes [11]. Various methods are employed to fabricate thin films of tungsten trioxide, such as hydrothermal techniques [12], solvothermal technique [13], pulsed laser deposition [1], radio-frequency magnetron sputtering [8], electrodeposition [14], atomic layer deposition [15], etc. In this study, Eu_2_O_3_-embedded WO_3_ films are synthesized using radio-frequency magnetron sputtering (RFMS), since it is a very simple technique, so by using this technique high-quality homogeneous films can be deposited.

Rare earth (RE)-based luminescence systems have gained considerable attraction due to their significant use in various technological fields. It is a well-established fact that the emission behavior of RE ions is greatly influenced by the surrounding environment of the host. Among RE elements, europium has a particular interest as a dopant, since it exhibits the property called valence fluctuation, i.e., it displays two valence states: +2 or +3. Depending on the valency, europium ions show different characteristic emissions [16]. Eu^3+^ ions exhibit an intense red emission via intra-4f transition, while Eu^2+^ ions present a broad range of emissions in the red to UV region due to 5d–4f transitions, which are influenced by the surrounding host matrix [16]. In europium-ion-doped host lattices, the intensity of emission peaks and their positions have a strong relationship with their morphology, grain size distribution and chemical composition, etc. [17]. In the present work, europium-embedded thin WO_3_ films are deposited using RF-magnetron sputtering, and their various properties are analyzed by adopting different characterization techniques.

## 2. Materials and Methods

The targets for sputtering were synthesized from WO_3_ and europium oxide (Eu_2_O_3_) (purity ~99.99%-Aldrich). Tungsten oxide powder was mixed with some selected contents (i.e., 0.0, 1.0, 3.0, 5.0 and 10.0 wt%) of Eu_2_O_3_ powder, then the mixture was ground, and a fine powder was obtained, which was taken as the target for film coating. Films were coated on amorphous quartz plate fixed at 50 mm distance w.r.t. target material. The details of film preparation have already been explained in our previously published work [18]. The sputter-coated films were then annealed at 600 °C (1 h). The annealed films, with different Eu_2_O_3_ contents, viz. 0.0, 1.0, 3.0, 5.0 and 10.0 wt%, are marked as WEu0, WEu1, WEu3, WEu5 and WEu10, respectively.

Structural studies and phase identification of the un-doped and Eu_2_O_3_-doped WO_3_ films were analyzed by studying the X-ray diffractograms, recorded with XPERT PRO X-ray diffractometer (Bruker). By employing Bragg–Brentano geometry, X-ray diffractograms were measured in 10–70° range (with an increment of ~0.0203° and scan speed ~2°/min) using CuK_α_ radiation of wavelength, λ = 1.5406 Å. Raman spectroscopic studies were done using Labram-HR800 micro-Raman spectrometer (HORIBA JobinYvon) fitted with an Ar-ion (λ = 514.5 nm) laser. Nova NanoSEM 450 (Field Emission) SEM was employed to analyze the topography of the prepared films. Moreover, atomic force microscopic (AFM) images were used to study the geometry of the synthesized thin films using Bruker Dimension Edge (Scan Asyst) atomic force microscope. Constituent elements in the fabricated thin films were analyzed using AMETEK EDAX Octane series attached to the Carl Zeiss EVO 18 scanning electron microscope. The ionized states of the respective elements were probed by using X-ray photoelectron spectroscopic (Thermo Scientific ESCALAB 250Xi) measurements. VeecoDektak 6M profilometer was employed to determine the films thickness. Transmittance and absorbance studies were analyzed using JASCO V-550 UV–VIS double-beam spectrophotometer with a spectral resolution of ~1 nm. The emission behavior of the films was studied using FLS980 spectrofluorometer (Edinburgh) provided with a 450 W xenon lamp (continuous).

## 3. Results and Discussion

### 3.1. Structural Analysis

The structural identification as well as the variation in the crystalline nature of the WO_3_ films with a change in Eu^3+^-doping concentrations are analyzed using the X-ray diffraction technique. Figure 1a–e presents the X-ray diffraction profiles of the Eu^3+^-doped WO_3_ thin films annealed at 600 °C. It is clear that all the films, except the WEu10 film, show a poly-crystalline structure and present the typical X-ray diffraction pattern of the monoclinic crystalline phase (ICDD Card No. 83–0951), while the XRD pattern of the WEu10 film does not present any XRD peaks, indicating its amorphous nature. In the diffraction patterns, peaks related to the dopant impurity are not detected at all. This result demonstrates that the dopant atoms are well-diffused without any segregation of the Eu_2_O_3_ phase in the deposited films. For all crystalline films, the XRD patterns show an a-axis orientation along the (*h*00) crystal plane despite, the added dopant, because of the reduced surface free energy quantity [19]. This strong orientation of the film along a selected crystal plane can be described using the model given by Van der Drift (“survival of the fastest” model) [20,21]. In this model, Van der Drift explained that at the initial film growth stage, nucleation takes place at different orientations, and these nuclei compete to enlarge in size. However, at the final stage of film growth, a nucleus having a larger growth rate exists, so the film will be oriented in that particular direction. From Figure 1f, it can be seen that the intensity of the peak along the (*h*00) crystal plane is enhanced by the increase in Eu^3+^-doping content up to 3 wt%; beyond that, the intensity reduces, and, finally, the film became amorphous at higher Eu^3+^ level (at 10 wt%). At low doping concentrations, the growth of the film along the a-axis can be attributed to the development of additional growth centers in the presence of Eu^3+^ ions, which may reduce the nucleation energy barrier [19,21].

Regarding the XRD peak, the full width at half maximum (FWHM) intensity for the intense (200) peak was estimated for the crystalline films, and the variation of the FWHM of the (200) peak with Eu^3+^ doping content is presented in Figure 1f and Table 1. The FWHM value for the (200) peak of pure film was found to be 0.2148°. With an increase in Eu^3+^-doping concentration, the FWHM decreases up to 3 wt% of Eu^3+^ concentration, and, after that, it increases. Out of the prepared films, the WEu3 film exhibits a low FWHM, indicating its superior crystalline quality among the films. From this observation, it can be established that moderate doping of the Eu^3+^ ions in the WO_3_ lattice enhances their crystalline nature, while a high concentration of Eu^3+^ ions decreases the crystalline behavior. The reason behind this reduction in crystallinity at a higher doping concentration can be understood from the following facts: (i) at a higher impurity concentration, the impurity atoms may segregate along the inter-particle boundary regions [22]; (ii) the addition of newer growth centers may become abstained at a higher impurity concentration [23]; (iii) the crystal reorientation effects become predominant at higher Eu^3+^ concentrations [24]. These factors can considerably reduce the crystalline quality of WO_3_ thin films with more Eu^3+^ contents. The mean crystallites size $\left(D_{hkl}\right)$ in the crystalline films is calculated from the $\left(2\theta_{hkl}\right)$ position and broadening ($\beta_{hkl}$, FWHM) of the most intense peak using the Scherrer formula (Equation (S1), additional information) [25]. The evaluated quantities of $\left(D_{hkl}\right)$ are shown in Table 1, which is in the range of 30–73 nm. For the pure film, the size of the mean crystallites is around 38 nm. In the lightly Eu^3+^-doped WO_3_ films, i.e., for WEu1 and WEu3 films, the mean crystallites size is 70 and 73 nm, respectively, whereas, in the WEu5 film, the average crystallites size reduces to 30 nm. This crystallite-size reduction in higher Eu^3+^-doped WO_3_ films may be because of the accumulation of the dopant atoms along the grain boundary region, which will produce a hindering force. When the hindering force exceeds the driving force for the grain growth, the movement of the grain boundary is curbed and, therefore, reduces the average size of crystallite [26].

The center of the (200) peak shows a systematic shift towards the lower angle side with the enhancement in Eu^3+^-doping concentration, and the observed change in the peak position is shown in Figure 2a. This regular variation in the 2*θ* value may be because of the longer ionic radius of the incorporated Eu^3+^ (~0.94 Å) [27] compared to the W^6+^ host ions (~0.65 Å) [28]. According to Vegard’s law [29], the incorporation of the impurity ions of the longer ionic radii into a lattice of the shorter ionic radii will expand the lattice due to substitutional incorporation, and, hence, the diffraction patterns will move to lower diffraction angles (2*θ*). The inter-planar distance of the (200) plane ($d_{200}$) is also calculated using Bragg’s relation, which is shown in Figure 2b and in Table 1. It is clear that the value of $d_{200}$ increases with a rise in Eu^3+^-doping concentrations. This variation of the most intense peak location towards a lower angle side and the marked increase in the value of $d_{200}$ indicate the WO_3_ lattice expansion, due to the presence of impurity ions. This expansion of the host lattice will produce stress in the thin films, which may arise due to different factors. The presence of defects and distortion and also the deviation in the ionic size of the host and impurity ions produce intrinsic stress in the deposited films. The disparity in the lattice constants and thermal expansion coefficients between the fabricated film and quartz substrate create extrinsic stress [30]. In the present study, due to the amorphous behavior of the quartz substrate, it is difficult to estimate the extrinsic stress that originated because of the lattice mismatch between the quartz substrate and thin film [31]. However, it was reported that this type of extrinsic stress might relax upon the nucleation of the particles, followed by a 3D structure formation on the 2D surface [32]. The role of temperature-dependent extrinsic stress can also be neglected when the film thickness is large [33]. In the present study, film’s thickness is determined with the help of stylus profilometry, which is in the region of 134–301 nm, as shown in Table 2; therefore, the effect due to thermal stress can be neglected. Hence, the stress in the thin films has a major role from the intrinsic portion, and this is mainly due to the ionic size variation between the added impurity and host ions. Moreover, the film-deposition conditions and the film’s surface topography have a unique role in the formation of stress.

The strain in the fabricated thin films is evaluated by the value of $\theta_{hkl}$ and $\beta_{hkl}$ for Bragg’s intense reflection plane, with the help of Appendix A [25]. Table 1 gives the determined quantities of the lattice strain. The dislocation density (δ), which is the length of the dislocation lines/unit volume, can be determined using Appendix A [34]. The estimated values of ‘δ’ are also shown in Table 1. It is clear that with the rise in Eu^3+^-doping concentrations, the dislocation density first decreases up to 3 wt% of Eu^3+^ level, and thereafter its value increases. This shows that the WEu3 film possesses the best crystalline property among the synthesized films [35].

Figure 3a–e show the micro-Raman spectra of the pure and Eu^3+^-embedded WO_3_ films measured (100–1100 cm^−1^) with an argon-ion laser (514.5 nm excitation). The literature shows that the phase of WO_3_ can be successfully identified using Raman spectroscopy [36,37]. The spectrum of each sample shown in the figure can be categorized into three fundamental vibrational groups. The stretching vibrational modes can be noticed in the 900–600 cm^−1^ range; the bending vibrational modes can be noticed in the 500–200 cm^−1^ range; and the lattice vibrational modes can be noticed in the wavenumber region below 200 cm^−1^ [38]. The spectra show a large number of vibrational modes, especially in the low wavenumber domain, which suggests a lowering of the symmetry and an enhanced number of molecules per unit cells [39]. The structure of WO_3_ is reported as distorted ReO_3_, and it contains corner-shared, distorted WO_6_ octahedra [40]. Group theory studies for an ideal ReO_3_ geometry (space group: $O_{h}^{1}$) reveal only two active Raman bands, but, in the present analysis, a higher number of active bands are detected in the spectra, probably because of the lowering of the symmetry and the distortion in the octahedra in the real monoclinic structure [37]. The Raman bands detected around 273, 322, 700 and 804 cm^−1^ are the characteristic Raman bands of the monoclinic WO_3_ phase, supporting the information obtained from XRD [41,42]. The intense band detected at 804 cm^−1^ is because of the symmetric stretching mode of O-W-O bonds, while those detected around 700 cm^−1^ are because of their asymmetric stretching vibrations. In higher Eu^3+^-embedded films, the symmetric stretching-mode position is observed at a lower wavenumber side compared to that in lightly doped films. The bands around 273 and 325 cm^−1^ can be attributed to the symmetric bending vibrations of the O-W-O bonds, while those between the 370 and 495 cm^−1^ regions can be ascribed to the asymmetric bending modes of the same bonds [36]. The FWHM (around ~804 cm^−1^) of the symmetric stretching vibrational mode is an indication of the structural disorder in terms of the bond length and angle of the W-O-W bond [43]. For the un-doped film, the FWHM value for this mode is around 34.148 cm^−1^. Upon increasing the Eu^3+^-doping concentration, the FWHM value first decreases up to 3 wt%, and thereafter it increases. The variation of the FWHM for the Eu^3+^-doping concentration is shown in Figure 3f. Out of the prepared films, the WEu3 film exhibits the lowest quantity (~21.945 cm^−1^) for FWHM, while theWEu10 film shows the highest value (~73.626 cm^−1^). This result is consistent with the XRD analysis: the crystalline quality enhances with an increase in Eu^3+^ content up to 3 wt%, and, after that, the crystalline quality deteriorates with an increase in Eu^3+^ content. The vibrational band noticed at 990 cm^−1^ in all the films could be assigned to the W^6+^=O stretching vibrations of the bridging oxygen, which may exist due to clusters and void structures present on the film surface [44].

In the spectrum of the WEu10 film, the integrated intensity of the bands, except for the one around 990 cm^−1^, reduces considerably compared to the other films. The increase in the Raman band intensity corresponds to the W^6+^=O stretching vibration points that this film may contain species other than the crystalline WO_3_ phase, i.e., the sub-stoichiometric WO_3_ phase [45]. Even though the X-ray diffractogram of the WEu10 film is devoid of the characteristic XRD peaks, its Raman spectrum shows noticeable Raman bands that correspond to the WO_3_ monoclinic phase, which may be due to the semi-crystalline trend of the WEu10 film. This result points to the conclusion that Raman spectroscopy is a powerful technique to analyze the phase of semi-crystalline materials where XRD fails.

### 3.2. Morphological and Composition Analysis 

Figure 4 presents the FESEM images of the pure and Eu^3+^-embedded WO_3_ thin films. The WEu0 film exhibits a smooth surface morphology, but, for the WEu1 film, a thick systematic distribution of the similarly sized smaller grains can be clearly noticed. The surface morphology of the WEu3 film presents a random distribution of the isolated bigger grains. It is also clear that these grains have a tendency to coalesce together to form bigger grains. The WEu5 film also exhibits almost the same surface morphology as the WEu3 film, but the grains are fewer in number and larger in size. An entirely different morphology can be observed in the WEu10 film. The film reveals a smooth surface with clusters of smaller grains distributed randomly over the surface. This surface morphology supports the amorphous/semi-crystalline nature of the WEu10 film, revealed by XRD and micro-Raman analysis.

The 3D AFM morphologies of the un-doped and Eu^3+^-embedded WO_3_ films are presented in Figure 5a–e. As seen from the morphology, the AFM images of the films also present a similar surface morphology as that of the FESEM images. The AFM micrograph of the un-doped film exhibits a smooth surface, but for the 1 wt% Eu^3+^-doped film, a thick distribution of smaller grains (uniform sizes) having well-defined boundaries can be observed. In the WEu3 film, a random distribution of bigger grains having clear grain boundaries can be observed. In the WEu5 film, the agglomerated bigger grains can be seen, which are much lower in number and distributed here and there on the film surface; in the WEu10 film, a smooth surface morphology can be observed. The root mean square (RMS) surface roughness of the fabricated films is calculated by WSxM 5.0 Develop 6.3 software [46] and is shown in Table 2 and Figure 5f. Among the prepared thin films, the WEu3 and WEu5 films show higher magnitudes of RMS surface roughness. Moreover, the WEu10 film presents the lowest value for RMS surface roughness among the doped films. It is quite interesting to observe that the grain size from the XRD analysis and morphology images (i.e., FESEM and AFM) differ greatly. This is because, in the XRD measurement, the size of the scattering domain is determined as the grain size (or crystallite size), but, in the morphology analysis, the surface morphology of agglomerated grains is measured, which mostly depends on the instrument’s resolution. Hence, it can be understood that an average grain observed in the morphology images carries other smaller crystallites that have individual orientations, which are detected using XRD [47].

To analyze the constituent elements in the synthesized thin films, the EDX spectra are taken. Figure 6 presents the EDX spectra of the pure and 3 wt% Eu^3+^-embedded WO_3_ films. For the spectrum of the un-doped film, we can only see the peaks related to ‘O’, ‘W’ and ‘Si’, but, for the 3 wt% Eu^3+^-embedded film, in addition to these peaks, the ‘Eu’ peaks can be clearly seen. The presence of the ‘Eu’ peaks in the embedded film indicates the presence of dopant atoms in the host lattice. The detected ‘Si’ peak originates from the quartz substrate used in the present study. It is important to note that in the EDX spectra of the films, we can observe an enhancement in intensity corresponding to the ‘W’ peak. This enhancement in the ‘W’ peak may be because of the overlap of the X-ray signals from the ‘W’ atom that is present in the deposited films with that of the emission from the ‘Si’ atom present in the quartz substrate [48]. The two-dimensional distribution of the different elements in the WEu5 film is given in Figure 7, and the mapping image confirms the incorporation of Eu^3+^ in the WO_3_ lattice. 

The XPS measurement analyzed the constituent elements and their respective oxidation states in the deposited films. The survey scan of the un-doped WO_3_ film (WEu0) (Figure 8a) presented peaks due to the ‘O’, ‘W’ and ‘C’ elements, but, for the Eu^3+^-embedded WO_3_ film (WEu3) (Figure 8b), the survey spectrum showed additional peaks corresponding to the ‘Eu’ element. The C1s peak observed at 284.98 eV might be because of the presence of the carbon impurity found on the surface of the film upon exposure to the surrounding atmosphere [49]. Figure 9a and Figure 9c, respectively, present the high-resolution XPS spectral profile of the WEu0 and WEu3 films in the W4f and W5p regions, along with their de-convoluted curves. In both the spectra, the 4f region contains a doublet, and the 5p region contains a singlet, corresponding to the ‘W’ atom. The doublet peaks in the 4f region of the WEu0 film are detected around 35.61 (4f_7/2_) and 37.59 eV (4f_5/2_), while, for the WEu3 film, these peaks are observed around 35.71 (4f_7/2_) and 37.69 eV (4f_5/2_). The singlet peak in the 5p region of the WEu0 film is observed around 41.34 eV, and, for the WEu3 film, it is observed around 41.28 eV. The binding energy (BE) values of the W4f_5/2_ and W4f_7/2_ peaks in these spectra show the +6 valence state of the ‘W’ atom in WO_3_ [50].

The O1s peaks of these films (WEu0 and WEu3) are presented in Figure 9b and Figure 9d, respectively. The asymmetric peak observed in both the spectra can be de-convoluted into two symmetric peaks; one at a low binding energy position, and the other at a high binding energy position. The peak observed at the low binding energy position (530.29 eV for WEu0 and 530.19 eV for WEu3) can be assigned to the O^2-^ ions bonded to the W^6+^ ions, and the peak observed at the high binding energy position (532.11 eV for WEu0 and 531.93 eV for WEu3) can be attributed to the adsorbed oxygen atoms on the film’s surface [51]. Figure 9e presents the high-resolution XPS spectrum of the Eu3d core energy level in the WEu3 film and its fitted curves. The spectrum presents two prominent peaks at 1135.10 and 1164.51 eV, which can be, respectively, attributed to the 3d_3/2_ and 3d_5/2_ spin-orbit splitting for the Eu^3+^ state [52]. Hence, it is quite evident that the dopant Eu exists primarily in the +3 states in the embedded films.

### 3.3. Optical Analysis 

The absorbance and transmittance spectra of the un-doped and Eu^3+^-embedded WO_3_ films in the 200–900 nm region are given in Figure 10a and Figure 10b, respectively. The transmittance of thin films mainly depends on a variety of parameters including crystalline quality, surface smoothness, thickness, porosity, etc. An increase in crystalline quality usually increases the optical transparency. In contrast, an increase in porosity, surface roughness and thickness decreases the optical transparency [53]. The prepared thin films show an average transmittance of ~51–85% in the 400–900 nm wavelength regions (Table 2). Among the prepared thin films, the WEu3 and WEu5 films exhibit lower values of optical transparency in the measured wavelength regions. Structural analysis shows an increased crystalline quality for the WEu3 film relative to the others; at the same time, this film has larger values for thickness and RMS roughness. This increased film thickness and surface roughness of the WEu3 film may be the reason for its reduced optical transmittance. However, for the WEu5 film, a reduction in crystallinity and increases in roughness and thickness have been observed. These reasons may be accounted for the reduced optical transparency of the WEu5 film. Moreover, the WEu10 film with the least crystalline quality may indicate that it possesses the lowest optical transparency among all the films, but, on the other hand, a good optical transparency has been noticed for this film. The reduced RMS surface roughness and film thickness showed by the WEu10 film may be the reason for its good optical transmittance.

The natures of the energy gap and optical transition of the prepared thin films are analyzed using the relation of photon energy (hν) and absorption coefficient ‘α’ using Appendix A [54]. In the present case, the $\alpha^{\frac{1}{2}}$ vs. hν plots (Figure 11a) show a linear nature, suggesting an indirect transition behavior for the deposited films. Table 2 shows the calculated quantities of the band gap energies. The variation of band gap energy with respect to the Eu^3+^-doping content is shown in Figure 11b. The band gap energy of the films is in the range of 2.49–2.95 eV. The un-doped WO_3_ film shows an energy gap of around 2.95 eV, but, for the Eu^3+^-doped films, the band gap energy gradually decreases with an increase in the doping content. The energy gap of a material depends on various factors such as defects, disorders, carrier concentration, strain, etc. [55,56]. For different RE-embedded CdO films, a red shift in the band gap energy was reported by Dakhel [57,58,59,60,61]. Dakhel suggested that the sudden increase in carrier concentration may be the possible reason for the decrease in the band gap energy with an increase in doping concentration, though this result is not in agreement with the expected band gap expansion because of the Burstein–Moss effect [62]. In semiconductors, the enhancement in carrier concentration produces two different effects, i.e., band gap shrinkage and band gap expansion.

The decrease in the energy gap may be due to the variation in the strength and nature of the crystalline potential by the incorporation of RE ions, which include the effect of their 4f shell valence electrons on the crystalline electronic energy levels [58]. Moreover, the increase in doping content increases the number of cations, which may enhance the localization of electrons that changes (increases) the concentration of the donor centers. This enhancement in donor concentration level effectively reduces the optical band gap [63]. The observed strain in thin films can also change the energy gap [64,65]: a compression strain increases the energy gap because of the compressed lattice, while a tensile strain reduces the energy gap due to the elongated lattice [55,64,65]. Structural analysis from the XRD measurement reveals an enhancement in the tensile strain because of the higher ionic radius of dopant (Eu^3+^) ions, and this increase in the tensile strain also decreases the band gap energy. For the WEu5 and WEu10 films, a reduction in crystalline quality is observed from the XRD and micro-Raman analysis. This reduction in crystallinity may produce oxygen vacancy sites, which may shrink the energy gap at a higher Eu^3+^ concentration due to the creation of deep localized states [48,66]. After all, the many body effects on the valence and conduction band reduce the width of the forbidden energy gap due to impurity scattering and electron interaction. Due to many body effects, the impurity levels merges with the up-lying conduction band and thus decreases the band gap energy [67].

The PL emission spectra of the un-doped and Eu^3+^-embedded WO_3_ films measured using 360 nm excitation are shown in Figure 12a. A few reports are available related to the emission properties of the Eu^3+^ doped WO_3_ film, but there are no reports analyzing the emission property of an RFM-sputtered WO_3_:Eu^3+^ thin film system. Luo et al. [68] reported an enhanced electrochromic switching behavior and a tunable red emission in response to an external bias from the Eu - doped WO_3_ films deposited via the hydrothermal process. Shen et al. [69] studied the effect of pH on the luminescent and electrochromic properties of an Eu-doped WO_3_ film prepared using the hydrothermal method and found that these properties could be changed by altering the pH value. Usually, for WO_3_, a wide emission spanning from NUV to the VIS spectral region (~390–500 nm) is reported, which relates to the indirect band to band transition in WO_3_ [70,71,72]. In the present work, un-doped WO_3_ film presents a wide emission peak in the 390–500 nm regions, because of the band-to-band transition in WO_3_. For Eu^3+^-doped films (except the WEu10 film), besides the band-to-band emission, emission peaks centered around 466, 537, 586, 614 and 686 nm regions are also observed. These peaks can, respectively, be assigned to the ^5^*D*_2_→^7^*F*_0_, ^5^*D*_1_→^7^*F*_1_, ^5^*D*_0_→^7^*F*_1_, →^7^*F*_2_ and →^7^*F*_4_ transitions in the europium ions having a valency of +3 (i.e., Eu^3+^) [73,74]. The emission around 614 nm (^5^*D*_0_→^7^*F*_2_) is a hypersensitive electric-dipole (ED) transition, and its intensity is highly affected by the surrounding crystal-field of Eu^3+^ ions. The radiative de-excitation due to ^5^*D*_0_→^7^*F*_1_ (586 nm) transition is a magnetic-dipole (MD) transition, which is independent of the surrounding crystal-field effect of the Eu^3+^ ions [75,76,77,78]. The intensity ratio of ED to MD transitions is commonly taken to analyze the chemical microenvironment of the Eu^3+^ ions in a given host [68]. Usually, the ED transition intensity becomes prominent in the PL spectra when the Eu^3+^ ions are situated at low symmetry sites having no inversion center. A strong red emission (~614 nm) in the WO_3_:Eu system reveals that Eu^3+^ occupies the non-inversion low symmetry centers in the WO_3_ host environment [74]. When the Eu^3+^ ion replaces W^6+^, the charge compensation will produce two types of defects: interstitial Eu^3+^ and oxygen vacancies. These defects may create symmetry distortion in the surrounding environment of Eu^3+^. Moreover, the disparity in the ionic radii of Eu^3+^ and W^6+^ produces a symmetry distortion in the surrounding host environment of Eu^3+^. This distortion in the local environment symmetry of the Eu^3+^ ions may enhance the ~614 nm emission intensity in the PL spectra of Eu^3+^-embedded WO_3_ films [79].

An enhancement in the emission intensity attributed to the band-to-band transition is observed with an enhancement in the Eu^3+^ content up to 3 wt%; after that, the intensity reduces with an enhancement in the Eu^3+^ content. This enhancement in excitonic emission intensity may be because of the enhanced crystalline quality of the films, up to 3 wt% of the Eu^3+^ content; beyond that, the decrease in the crystalline property of the films leads to a reduction in the emission intensity of the excitonic transition [18]. Moreover, the superposition of the characteristic emissions of europium ions with that of the band edge emission may also contribute to an increased excitonic emission intensity, up to 3 wt% of the Eu^3+^ion concentration in the doped films. Moreover, the overall enhancement in the emission intensities of the films, upto 3 wt% of the Eu^3+^ concentration, can also be related to the roughness values of the films. Usually, a film having a high surface roughness shows an enhanced emission intensity, which can be correlated to the extraction of a strong emission due to the increased light scattering from the walls of the crystallites. In this report, the high value of RMS roughness and the improved emission property shown by the WEu3 films may be because of the increased number of grains in them (Figure 4). The PL emission intensities of the films are in accordance with the values of RMS roughness.

It is also observed that the characteristic emission intensity of the europium ions decreases beyond 3 wt% of the Eu^3+^ concentration. This reduction in the characteristic emission intensity of the Eu^3+^ ions may be due to the well-known concentration quenching effect, because of the cross-relaxation occurring between neighboring Eu^3+^ ions [74]. At a low concentration of europium ions, the mutual interaction between neighboring Eu^3+^ ions can be neglected, but, at a higher concentration of dopant ions, the interaction between neighboring ions increases. Due to this mutual interaction, the energy of activated Eu^3+^ ions can be easily transferred to quenching centers to be released as non-radiative energy, which may result in the quenching of the characteristic emission intensity of the Eu^3+^ ions at higher Eu^3+^ ion concentrations [68]. It is very interesting to observe that the overall emission intensity for the WEu10 film is very much reduced compared to all other Eu^3+^-doped films. This observed reduction in the total emission intensity of the WEu10 film may be attributed to its amorphous/semi-crystalline nature, reduced film thickness and the enhanced concentration quenching effect, due to the high concentration of Eu^3+^ ions. The CIE-1931 chromaticity diagram is plotted to study the emission-color perception of the synthesized thin films [80] and is shown in Figure 12b. The estimated quantities of the CIE co-ordinates (x, y) are also shown in Table 2. As seen from Table 2 and Figure 12b, the color can be tuned from blue to bluish red by changing the Eu^3+^ ion concentration [81]. Moreover, both the WEu3 and WEu5 films exhibit more or less similar bluish-red color coordinates, whereas the WEu10 film generates close to a blue color region. For a higher europium ion concentration (WEu10), the host emission dominates, but the active ion emission has been reduced due to the high concentration quenching of the Eu^3+^ ions.

## 4. Conclusions

Eu^3+^-doped WO_3_ thin films are successfully deposited through the RF-magnetron sputtering method. An XRD study reveals that the crystalline property of the films is enhanced with an enhancement in the Eu^3+^ doping content up to 3 wt%; beyond that, a considerable reduction in crystallinity is observed. At a higher Eu^3+^ concentration (10 wt%), the XRD pattern reveals an amorphous nature. The XRD peaks in the crystalline films are indexed to the monoclinic crystal geometry of the WO_3_. The micro-Raman spectra of all the films show vibrational bands corresponding to the monoclinic crystal geometry of WO_3_. Even though the XRD pattern of the 10 wt% Eu^3+^-doped WO_3_ film is devoid of the characteristic XRD peaks, noticeable vibrational bands of monoclinic WO_3_ phase are observed in its Raman spectrum, indicating its semi-crystalline nature. The tendency of smaller grains to coalesce together to form bigger ones at a higher Eu^3+^-doping concentration can be observed from the analysis of the surface morphology of the films using FESEM and AFM micrographs. The EDX and XPS analyses indicate the incorporation of Eu^3+^ in the WO_3_ lattice. XPS analysis confirms the +3 oxidation state of the Eu in the doped samples. Optical study shows that the prepared thin films possess an average transmittance of ~51–85% in the 400–900 nm range. The band gap energy shows a reduction from 2.95 to 2.49 eV with an enhancement in the Eu^3+^ doping concentration. In addition to the band edge emission of WO_3_, the doped films (except the 10 wt% Eu^3+^-doped WO_3_ film) exhibit the characteristic emissions of the Eu^3+^ ions, due to the 4f intra configuration transitions in the Eu^3+^ ions. The reduction in the intensity of the characteristic emissions at a higher Eu^3+^-doping concentration can be ascribed to the concentration quenching effect, due to the cross-relaxation between neighboring europium ions. The CIE-1931 plot shows that the emission spectral range can be turned from blue to bluish red by changing the Eu^3+^ ion content.

## Figures and Tables

**Figure 1 nanomaterials-13-00007-f001:**
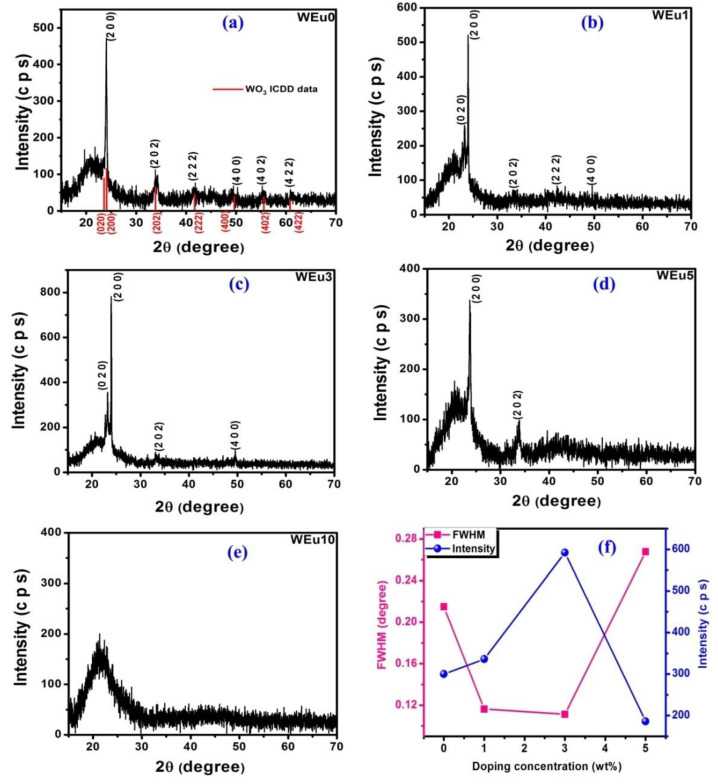
(**a**–**e**) XRD patterns of the un-doped and Eu^3+^-embedded WO_3_ films annealed at 600 °C; (**f**) variation of FWHM and intensity of the (200) peak as a function of Eu^3+^-doping content.

**Figure 2 nanomaterials-13-00007-f002:**
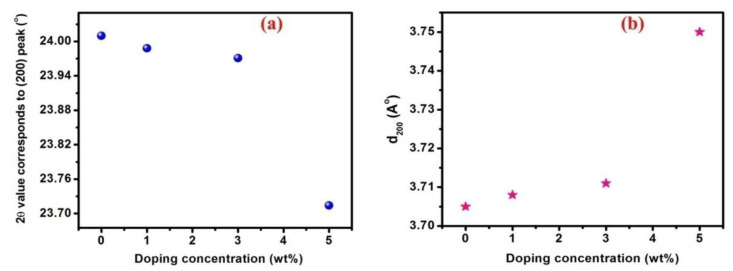
(**a**) Variation in the diffraction angle (2*θ*) (blue dots) and (**b**) inter-planar separation ($d_{200}$) (pink pentagrams) of the intense (200) peak as a function of Eu^3+^ content.

**Figure 3 nanomaterials-13-00007-f003:**
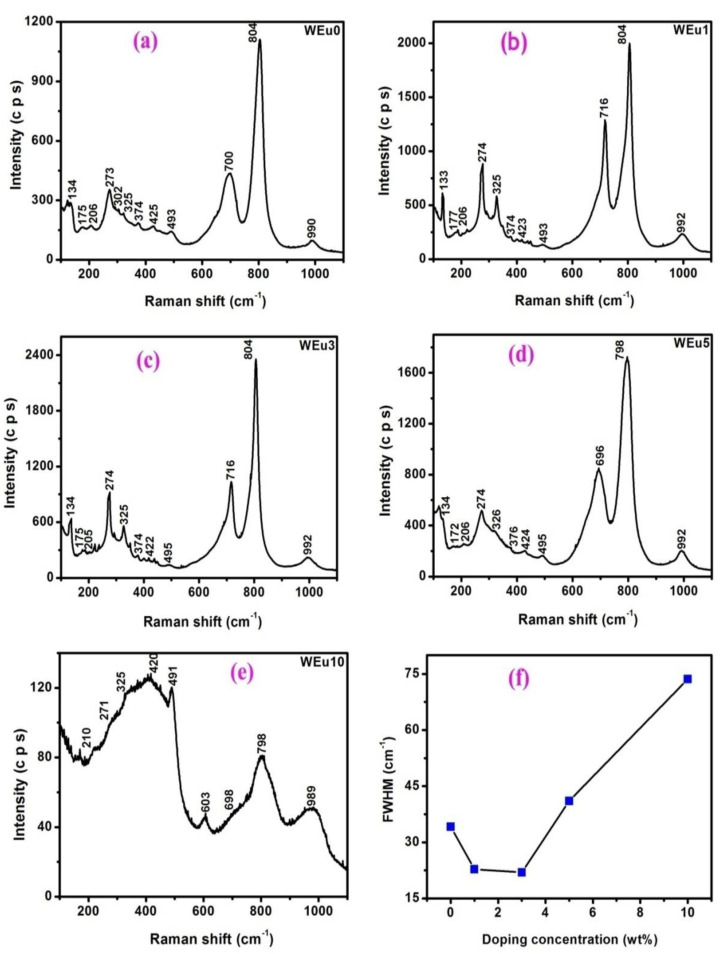
(**a**–**e**) Micro-Raman spectra of the un-doped and Eu^3+^-embedded WO_3_ films annealed at 600 °C; (**f**) change in FWHM of the symmetric stretching mode (around ~804 cm^−1^) as a function of Eu^3+^-doping content.

**Figure 4 nanomaterials-13-00007-f004:**
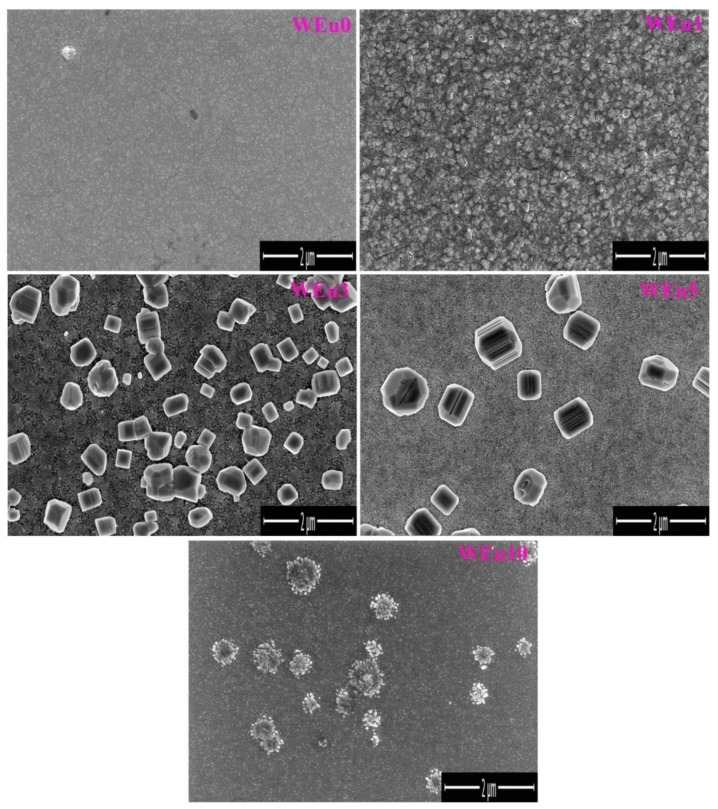
FESEM images of the un-doped and Eu^3+^-embedded WO_3_ films annealed at a temperature of 600 °C.

**Figure 5 nanomaterials-13-00007-f005:**
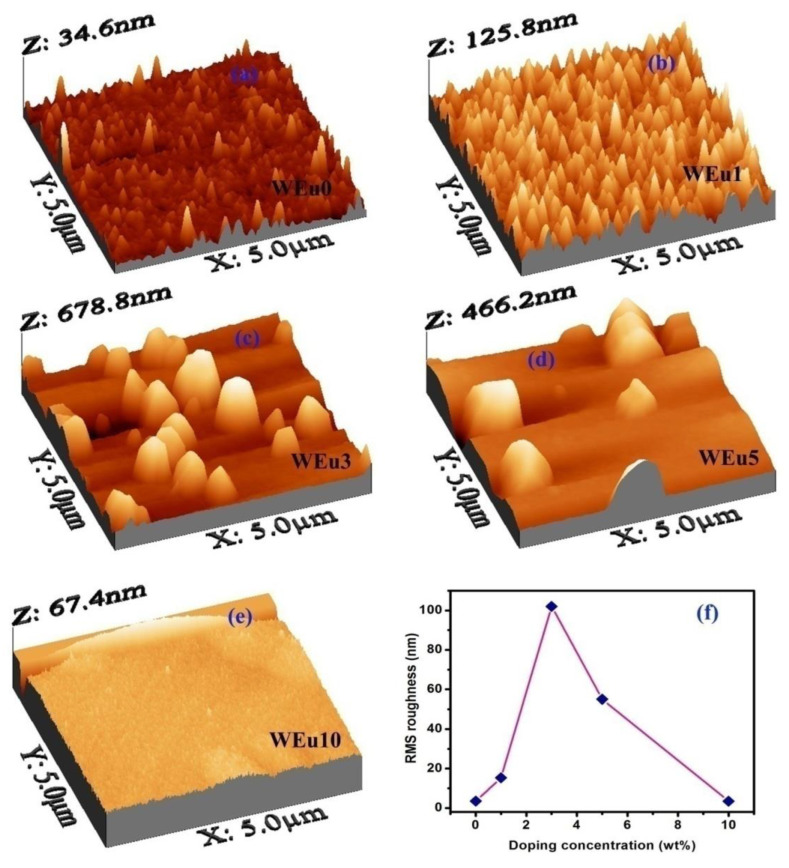
(**a**–**e**) Three-dimensional AFM images of the un-doped and Eu^3+^-embedded WO_3_ films annealed at a temperature of 600 °C; (**f**) variation of RMS surface roughness with respect to Eu^3+^-doping concentration.

**Figure 6 nanomaterials-13-00007-f006:**
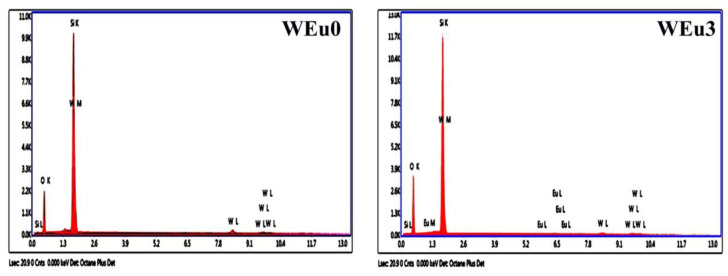
EDX spectra of the un-doped (WEu0) and 3 wt% Eu^3+^-embedded (WEu3) WO_3_ films annealed at a temperature of 600 °C.

**Figure 7 nanomaterials-13-00007-f007:**
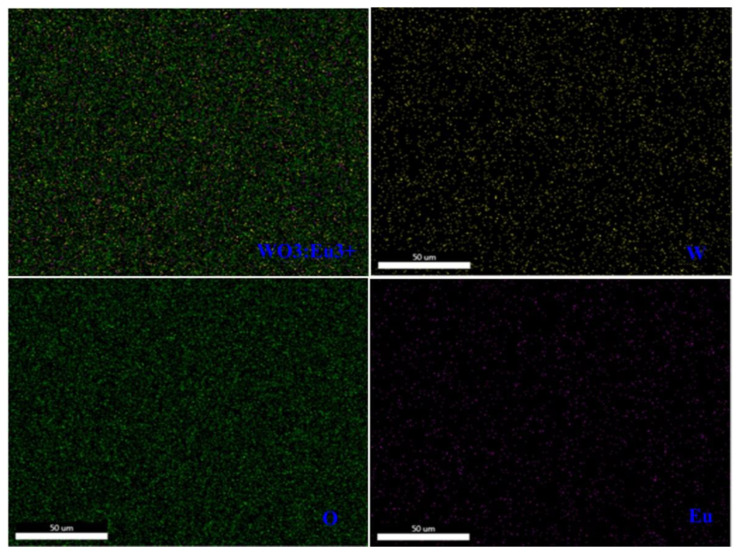
X-ray dot mapping image of WEu5 film.

**Figure 8 nanomaterials-13-00007-f008:**
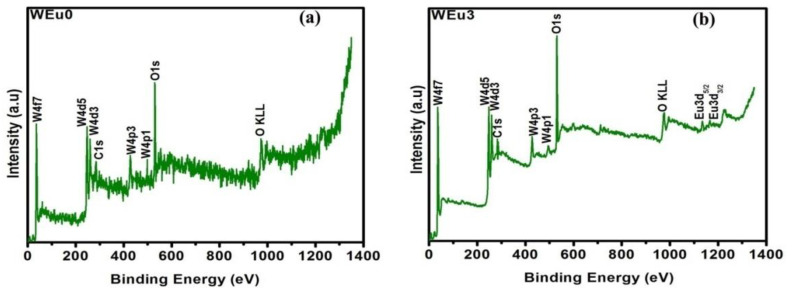
XPS survey spectra of (**a**) un-doped (WEu0) and (**b**) 3 wt% Eu^3+^-doped (WEu3) WO_3_ films annealed at 600°C.

**Figure 9 nanomaterials-13-00007-f009:**
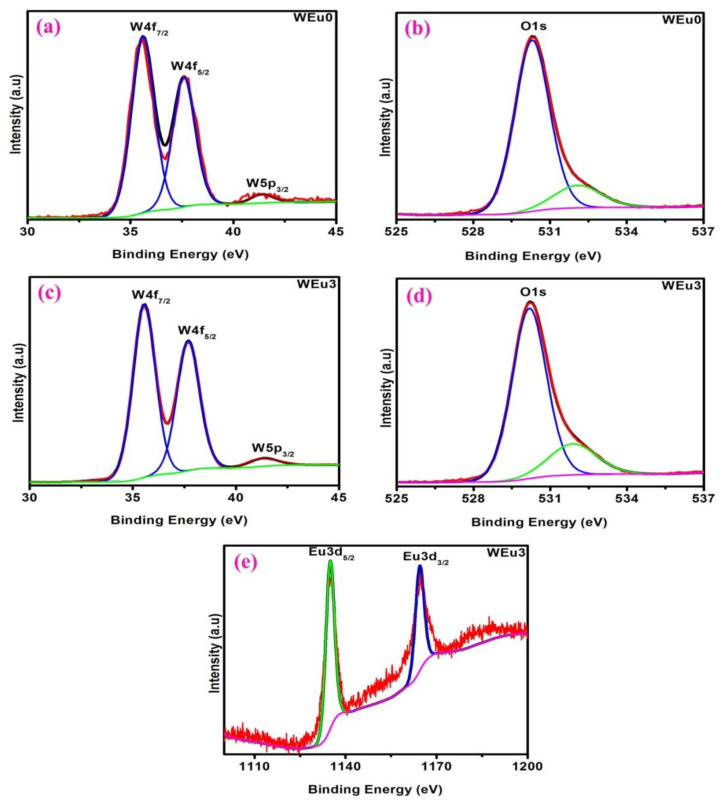
(**a**,**c**) Fitted W4f and W5p core energy level spectra of un-doped and 3 wt% Eu^3+^-embedded WO_3_ films; (**b**,**d**) fitted O1s core energy level spectra of un-doped and 3 wt% Eu^3+^-embedded WO_3_ films; (**e**) fitted Eu3d core energy level spectra of 3 wt% Eu^3+^-embedded WO_3_ film (In the figures; red colour—experimental curve; other colours—fitted curves).

**Figure 10 nanomaterials-13-00007-f010:**
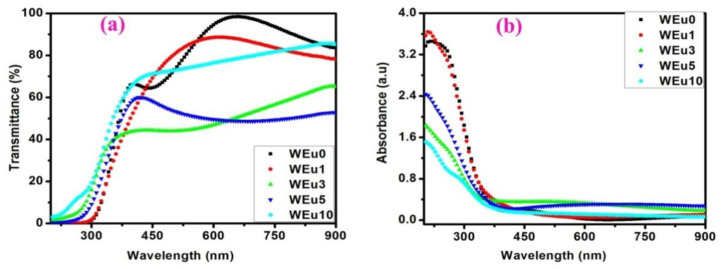
(**a**,**b**) Transmittance and absorbance spectra of the un-doped and Eu^3+^-embedded WO_3_ films annealed at 600 °C.

**Figure 11 nanomaterials-13-00007-f011:**
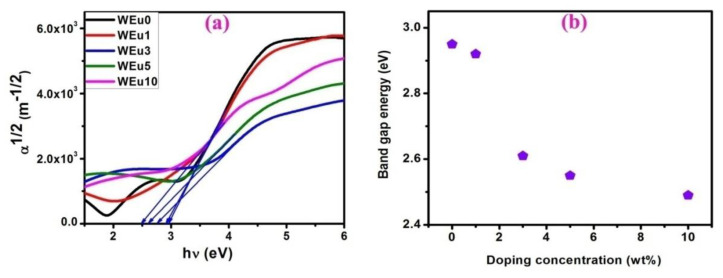
(**a**) $\alpha^{\frac{1}{2}}$ vs. *hν* graphs of the un-doped and Eu^3+^-embedded WO_3_ films; (**b**) change of band gap energy with respect to Eu^3+^-doping content.

**Figure 12 nanomaterials-13-00007-f012:**
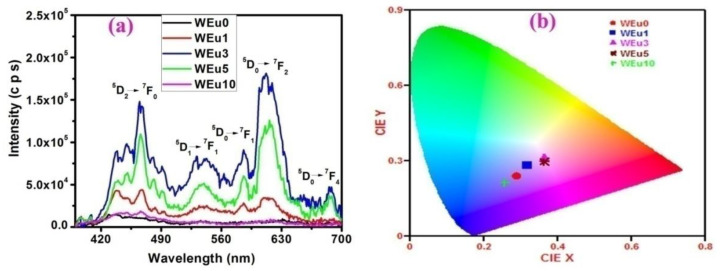
(**a**) PL emission spectra and (**b**) CIE chromaticity plot of the un-doped and Eu^3+^-embedded WO_3_ films annealed at 600 °C.

**Table 1 nanomaterials-13-00007-t001:** Structural parameters of the un-doped and Eu^3+^-embedded WO_3_ films annealed at 600 °C.

Sample Code	FWHM (Degree)	Crystallite Mean Size (nm) from Scherrer Formula	Micro Strain (×10^−3^)	Dislocation Density (lines/nm^2^)	$d_{200}$ (nm)
WEu0	0.2148	38	1.97	0.00069	0.3705
WEu1	0.1162	70	1.06	0.00020	0.3708
WEu3	0.1112	73	1.03	0.00018	0.3711
WEu5	0.2678	30	2.50	0.0011	0.3750

**Table 2 nanomaterials-13-00007-t002:** Optical and morphological parameters of the un-doped and Eu^3+^-embedded WO_3_ films annealed at 600 °C.

Sample Label	Film Thickness (nm)	RMS Surface Roughness (nm)	Average Transmittance (%)	Band Gap Energy E_g_(eV)	CIE Coordinates	
					X	Y
WEu0	243	3.48	85	2.95	0.2893	0.2419
WEu1	251	15.29	80	2.92	0.3180	0.2826
WEu3	280	102.01	51	2.61	0.3635	0.3111
WEu5	301	55.07	51	2.55	0.3624	0.2961
WEu10	134	3.49	78	2.49	0.2547	0.2126

## Data Availability

Not applicable.

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
