# Peer review of "Tailoring the Emission Behavior of WO3 Thin Films by Eu3+ Ions for Light-Emitting Applications"

_nanomaterials, 2022, doi:10.3390/nano13010007_

Round 1

Reviewer 1 Report

The manuscript shows structural and optical properties obtained for  Eu-doped WO3 thin films.

The introduction properly presents presumptions. Results are clearly presented. Figures are sufficiently communicative. The comparison of results to the literature data is correct. The conclusions are well formulated. The reference list is adequate.

The presented results are valuable for publication. 

I recommend publishing the manuscript as it is.

Author Response

We thank the reviewer for his kind recommendation to publish the paper as it is.

Reviewer 2 Report

Kavitha et. al. reports the fabrication of different concentrations of Eu3+-doped WO3 thin films via radio frequency magnetron sputtering technique. In this work, the author found that the overall emission color can be tuned from blue to bluish-red by varying the Eu3+ ion concentration. This work is interesting. However, there are still many issues should be addressed before publication. I suggest the current manuscript under a minor-revision state.

1.    The abstract of this manuscript should be revised. The novelty of this work should be pointed out in the abstract instead of just description.

2.    In the Introduction section, or the main text of the manuscript, I suggest the author compare the result of this work with the previously published work.

3.   We can see the surface morphology and the grains size of WEu3 is between WEu1 and WEu5 in Figure 4. However, the impact on performance between the relevance of surface morphology and the grains size isn't explained.

4.   The EDX spectrogram is blurring in Figure 6. Therefore, the comparison between WEu0 and WEu3 can't be convincing.

Author Response

Dear Reviewer 2,

Thank you very much for your comments to complete and improve our manuscript. The response to reviewer has been decleared as following;

Point1. The abstract of this manuscript should be revised. The novelty of this work should be pointed out in the abstract instead of just description.

Response1: Thank you for the suggestion. As per reviewer comments we modified the text (Page no.1 in the revised manuscript).

Point2.  In the Introduction section, or the main text of the manuscript, I suggest the author compare the result of this work with the previously published work.

Response2: As per reviewer comment, suggested modifications are done in the manuscript (Page no.14 in the revised manuscript).

       Point3. We can see the surface morphology and the grains size of WEu3 is between WEu1 and WEu5 in Figure 4. However, the impact on performance between the relevance of surface morphology and the grains size isn't explained.

Response3: The morphology analysis using FESEM micrographs show larger grains for WEu3 and WEu5 films compared to other films. WEu3 film presents a random distribution of a large number of isolated bigger grains. It is also clear that these grains have a tendency to coalesce together to form bigger grains. WEu5 film also presents almost similar surface morphology as that of WEu3 film, but the grains are fewer in number and larger in size. The RMS surface roughness values of the deposited films are calculated from the AFM pictures and the highest value is obtained for WEu3 film due to the presence of large number of bigger grains. Usually films with high surface roughness show high PL emission intensity and in the present report the PL emission intensity is in accordance with the value of RMS surface roughness of the deposited films. Among the deposited films high emission intensity is obtained for WEu3 films may be because of the increased number of bigger grains in them. The PL enhancement can be correlated to the extraction of strong emission due to the increased light scattering from the walls of the crystallites.

In conjunction with the above explanation, the text in the manuscript is modified (Page no.15 in the revised manuscript)

Point4. The EDX spectrogram is blurring in Figure 6. Therefore, the comparison between WEu0 and WEu3 can't be convincing.

Response4: Sorry, these figures are system generated. However we tried to improve the quality of this figure.

Regards,

Authors

Reviewer 3 Report

This article reports the successful fabrication of Eu3+ doped WO3 thin films via radio frequency magnetron sputtering technique. Structural and morphologic analysis confirms the procedure to annealed Eu3+ doped WO3 thin films. The obtained hollow Eu3+ -doped WO3 films have great optical properties. The context of the article is clear. However, the characterizations for coatings are defective. Therefore, major revision is recommended. More detailed comments have been listed in below:

1. Whether the Eu3+ -doped WO3 film will have better optical properties without annealing. How to identify the WO3 phase for Eu3+ sample by the XRD and Micro-Raman pattern? The diffraction peaks of standard cards should be shown in the XRD pattern.

2. The valence changes of Eu3+ before and after heat treatment was not reflected. XPS should be performed to characterize the valence changes of different doping concentration Eu3+ before and after thermal treatment.

3. The film uniformity and element distribution can be more clearly characterized by various microscopes and element scanning means. High resolution TEM should be provided to characterize the crystal lattice.

Author Response

Dear Reviewer 3,

Thank you very much for your comments to complete and improve our manuscript. The response to reviewer has been decleared as following;

Point1. Whether the Eu3+ -doped WO3 film will have better optical properties without annealing. How to identify the WO3 phase for Eu3+ sample by the XRD and Micro-Raman pattern? The diffraction peaks of standard cards should be shown in the XRD pattern.

Response1: The as-deposited films are in amorphous state so it possesses reduced optical quality compared to the annealed samples.

In the diffraction patterns of the doped films, peaks related to the dopant impurity are not at all detected. This result point that the dopant atoms are well-diffused in the host matrix and there is no phase change due to impurity doping. So by comparing the XRD peak positions in the doped samples with the standard ICDD card used for the undoped sample we can find the phase of Eu3+ doped WO3 samples. Also, by comparing the obtained vibrational Raman bands for the doped samples with the vibrational bands obtained for a monoclinic WO3 phase (Report shows that the Raman bands positioned around 273, 322, 700 and 804 cm-1 are the typical Raman bands of monoclinic WO3 phase) we can find the phase of Eu3+ doped WO3 films.

Suggested modifications are done in the XRD pattern.

Point2. The valence changes of Eu3+ before and after heat treatment was not reflected. XPS should be performed to characterize the valence changes of different doping concentration Eu3+ before and after thermal treatment.

Response2: Thank you for the valuable suggestion. Unfortunately our RF-magnetron sputtering instrument is not functioning properly. It has some major problem so it will take some time to solve the issue. So we don’t have any facility to do the experiment right now. Kindly accept this.

Point3. The film uniformity and element distribution can be more clearly characterized by various microscopes and element scanning means. High resolution TEM should be provided to characterize the crystal lattice.

Response3: Thank you for the valuable suggestions. Sorry to say that we have only Field Emission Scanning Electron Microscope and Atomic Force microscope to study the morphology of the deposited samples and we have provided the images in the manuscript. The elemental mapping for the WEu5 sample is done and is given in the revised manuscript. HR-TEM instrument is not available either with us or nearby research centers. So, we are not able to do the HR-TEM analysis right now. Kindly accept this.

Regards,

Authors

Reviewer 4 Report

The authors of the manuscript successfully fabricated a set of Eu3+ doped WO3 thin films. The films were deposited on an amorphous quartz plate by radio frequency magnetron sputtering technique. This simple technique allows to deposit the high-quality homogeneous films. The obtained samples with different doping levels were studied in detail using the following instrumental methods: the X-ray diffraction and X-ray photoelectron spectroscopies, the photoluminescent and Raman spectroscopies, the transmission and absorption optical spectroscopies, the scanning electron and atomic force microscopies, as well the secondary electron microscopy.

Among the results obtained, it should be especially noted that: the thin films have the best crystallinity at an Eu3+ concentration of about 3 wt%; the thin films possess an average transmittance of ~51-85 % in the 400-900 nm region; an emission colour of the films can be tuned from blue to bluish-red by changing the Eu3+ ion concentration.

I believe that the proposed article will be of interest to a wide range of researchers working in the field of nanostructure physics and engineering. In my opinion, the manuscript can be published in Nanomaterials without essential changes. It is only required execute minor spell check.

Author Response

Dear Reviewer 4,

We thank the reviewer for his kind recommendation to publish the paper as it is after final spell check. We carefully re-checked and corrected the paper.

Regards,

Authors

Round 2

Reviewer 3 Report

The revised manuscript can be accepted.